# In situ nanoscale imaging of moiré superlattices in twisted van der Waals heterostructures

Yue Luo [1,2], Rebecca Engelke[2], Marios Mattheakis [3], Michele Tamagnone [3], Stephen Carr[2], Kenji Watanabe [4], Takashi Taniguchi [4], Efthimios Kaxiras[2,3], Philip Kim [2] & William L. Wilson [1✉]

Direct visualization of nanometer-scale properties of moiré superlattices in van der Waals heterostructure devices is a critically needed diagnostic tool for study of the electronic and optical phenomena induced by the periodic variation of atomic structure in these complex systems. Conventional imaging methods are destructive and insensitive to the buried device geometries, preventing practical inspection. Here we report a versatile scanning probe microscopy employing infrared light for imaging moiré superlattices of twisted bilayers graphene encapsulated by hexagonal boron nitride. We map the pattern using the scattering dynamics of phonon polaritons launched in hexagonal boron nitride capping layers via its interaction with the buried moiré superlattices. We explore the origin of the double-line features imaged and show the mechanism of the underlying effective phase change of the phonon polariton reflectance at domain walls. The nano-imaging tool developed provides a non-destructive analytical approach to elucidate the complex physics of moiré engineered heterostructures.

[1] Center for Nanoscale Systems, Harvard University, Cambridge, MA 02138, USA. [2] Department of Physics, Harvard University, Cambridge, MA 02138, USA. [3] John A. Paulson School of Engineering and Applied Science, Harvard University, Cambridge, MA 02138, USA. [4] National Institute for Materials Science, Namiki 1-1, Ibaraki 305-0044, Japan. ✉email: wwilson@cns.fas.harvard.edu

Complex functional materials and devices engineered via the interlayer stacking of two-dimensional (2D) van der Waals (vdW) layers[1–4], enable a unique control and design approach to novel heterostructures with optimal electronic and optical properties for advanced applications[2,5,6]. The periodic modulation of the local atomic registry by rotation of the vdW layers offers an additional degree of freedom resulting in moiré superlattices[7]. In particular, twisted bilayer graphene (TBG) exhibits strong electron correlations which lead to unconventional superconductivity at a "magic angle" of $\theta = 1.1°$[1]. Moreover the electronic band structures evolved via this platform at even lower angles (<0.5°) can lead to features with fascinating physical properties such as topologically protected quantum valley Hall edge states[8,9], moiré domain reconstruction[10] and topological transport in network of domain boundaries[11].

Several types of imaging techniques have been utilized to visualize moiré superlattices with high resolution such as transmission electron microscopy (TEM)[10,12] and scanning tunneling microscopy (STM)[13]. TEM direct imaging of the lattice structure is destructive, requiring samples to be placed onto specialized thin TEM grids which cannot be used for further device fabrication, (i.e., adding contacts for transport measurements, etc.) STM imaging of these systems requires samples to be conductive, therefore no hexagonal boron nitride (h-BN) encapsulation, (used to ensure environmental stability), can be used on top. In addition, ultra-high vacuum and/or cryogenic temperatures are needed for the lattice imaging. Moiré superlattices have also been imaged using atomic force microscopy (AFM) operated in the piezoresponse force microscopy mode, here sub-5 nm resolution has been achieved[14]. However, this method, like STM, still requires the AFM tip in direct contact with the region of interest of the heterostructure, thereby preventing any top protecting layer encapsulation or top gates. More recently infrared (IR) nano-imaging and spectroscopy using an aperture-less scattering-type scanning near-field optical microscope (s-SNOM) has been demonstrated as a powerful tool to investigate the spectral and quasiparticle dynamics of 2D materials beyond the diffraction limit[5,15]. Local optical conductivity changes across the domain walls allow domain wall networks to be directly visualized optically through s-SNOM[8,16].

Moiré patterns in TBG have recently been imaged by probing the propagation of the surface plasmon polariton (SPP) launched in doped graphene[17]. To enable this imaging approach, doping of the heterostructure through substrate treatment or electrostatic gating is required to change the Fermi level ($E_F$) in graphene increasing the density of free carriers (typical $E_F \geq 300$ meV)[17–19], moreover only an extremely thin top h-BN layer (<4 nm) is allowed in this case to avoid reducing the interaction strength between the AFM tip and the TBG, thus not an ideal method to characterize moiré superlattices in devices of interest. Unlike SPP in doped graphene, tunable phonon polaritons in h-BN are collective modes in this polar crystal where photons are strongly coupled to optical phonons, which have long propagation length without the need for electronic doping[5,20]. Control of these hyperbolic phonon polariton modes has been achieved by composing the vdW heterostructure of h-BN with a monolayer of graphene forming hybridized modes[18]. However, the hybrid modes still require electric doping in graphene. Alternatively, the electrodynamic properties of the phonon polariton modes can be modified by varying the elastic strain in the h-BN soliton superlattices[21]. This modification relies on aligning h-BN to the underneath layer thus hard to dynamically control. To overcome these limitations, we directly launch the tunable phonon polariton in the h-BN layers that are used to encapsulate the TBG and visualize the moiré superlattices in TBG via the modulation of the optical conductivity due to the reflective scattering of the quasiparticle at the domain walls of the buried heterostructure. We further explore this unusual reflection behavior of the phonon polariton at domain walls utilizing finite-difference time domain (FDTD) simulations.

## Results

**Nano-imaging of TBG superlattices.** The conventional s-SNOM geometry is illustrated in Fig. 1a, here a metal coated tip of an AFM

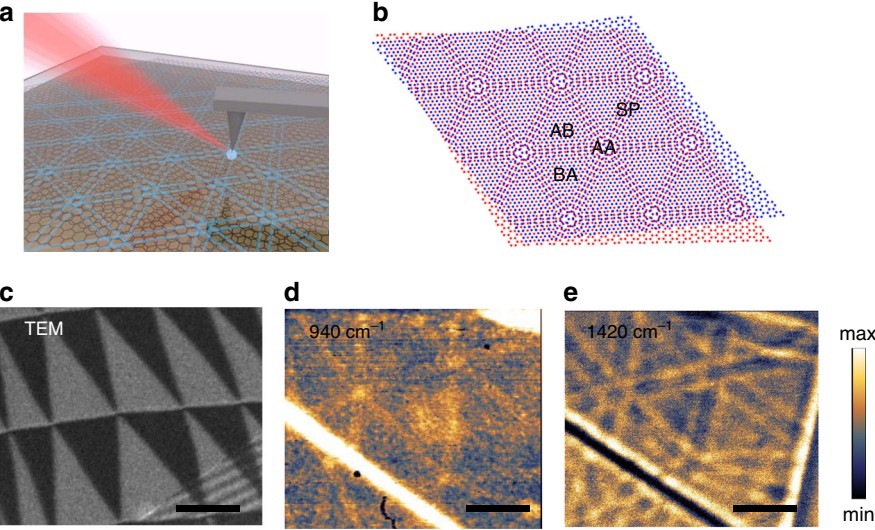

**Fig. 1 Overview of near-field nano-imaging of the hybridized response in twisted bilayer graphene heterostructure. a** Experimental schematic showing infrared light beams incident and scattered by an atomic force microscope tip. The near-field response is detected through the scattered light in the far-field. **b** Schematic diagram of twisted bilayer graphene showing different stacking configurations. **c** Transmission electron microscopy (TEM) dark-field image obtained by selecting the graphene diffraction peak in a TBG. The contrast of AB/BA domains is associated with the antisymmetric shift of lattice period in AB and BA domains. **d** and **e** Near-field images of the normalized near-field scattering amplitude $s(\omega)$ in false color maps revealing the domain walls in TBG without bottom h-BN taken at 940 cm$^{-1}$ (**d** the plasmon polariton resonant wavelength) and 1420 cm$^{-1}$ (**e** the phonon polariton resonant wavelength). The twisted angle is about 0.02°. The contrast seen of the domain walls is associated with the reflectivity of the plasmon polariton in graphene and phonon polariton in h-BN, respectively. Scale bar: 500 nm. Note: this sample was purposely doped to increase the plasmonic response.

is illuminated with a mid-IR beam generated by, (in our system), a quantum cascade laser source. The electric field between the tip and the sample is locally, strongly enhanced due to the small radius of the metal coated probe. This interaction can readily launch plasmon polariton waves along a graphene surface and/or phonon polariton waves along an h-BN surface layer when the optical field is tuned to the appropriate mid IR frequency. The collective quasiparticle excitations of this coupled system are then scattered into the far-field where they are detected by a photodetector and demodulated with a lock-in amplifier using the tapping frequency of the tip. Using a pseudo-heterodyne scheme, the amplitude and phase of the nearfield scattering can be extracted from the detected harmonics. Exploiting this technique to gain a deeper understanding of 2D heterostructure materials and their devices is an important new area of study[15,17,18,22]. Early work has utilized s-SNOM to directly image surfaces of 2D systems, mapping the quasiparticle dynamics in an array of 2D materials; here we show that one can exploit the interaction between layers to image the complex structure and quasiparticle dynamics of buried surfaces. This advance offers the ability to image the electronic landscape of pre and post processed devices undertest.

Engineered 2D heterostructure devices made from vdW material systems offer a new design space in the development of quantum devices and systems. One of the simplest heterostructure prototypes is the TBG structure encapsulated with h-BN thin layers. (h-BN is the most commonly used protective layers, due to its large bandgap). This system is an ideal platform to study the physics of the quasiparticles at the complex 2D domain boundaries where the moiré structure changes to the electronic structure can potentially be manipulated. For a pair of stacked graphene layers, there are three different types of symmetric stacking configurations revealed spatially driven by the possible vertical overlay of the two sublattices: the energetically favorable AB (BA) stacking, unfavorable AA stacking and saddle point stacking (Fig. 1b). In this study, we consider the small twist angle regime where atomic scale reconstruction at vdW interfaces induces a long length scale moiré pattern, forming well-aligned AB and BA domains separated by domain walls[23,24] as shown by the TEM dark-field (DF) imaging (Fig. 1c). The periodic array of sixfold sharp domain boundaries between the AB and BA domains is the result of the interfacial atomic reconstruction in TBG[25,26]. The same sixfold pattern can be observed by the near-field IR response when the IR light at frequency $\omega = 940\,\mathrm{cm}^{-1}$ is used to launch the plasmon polariton directly in the TBG layer (Fig. 1d). The bright lines along the solitons between the AB and BA domains reflect the local variation of the optical conductivity. As noted, utilizing s-SNOM here to image this feature using near-field signals from plasmon polariton in TBG requires heavy electrostatic gating or an oxygen plasma treatment of the substrate to increase the carrier density. In addition, because of the near-field interaction range, the top h-BN layer cannot be thick, which is not ideal for protection in any practical device application. In contrast, as shown in Fig. 1e, by tuning the excitation laser frequency $\omega = 1420\,\mathrm{cm}^{-1}$ resonant with the upper reststrahlen band (RB) region of phonon polariton in h-BN, (where dielectric functions $\varepsilon_z > 0$, $\varepsilon_{xy} < 0$, spanning the range $\omega = 1370–1610\,\mathrm{cm}^{-1}$), we can observe the sixfold domain wall arrays having the same periods of moiré patterns in terms of the domain size, important here is this illustrates we can clearly observe the substructure of the buried TBG device even though it is fully encapsulated, protected by h-BN with both top and bottom layers with thickness of 8–16 nm and ~26 nm, respectively.

**Elucidating phonon polariton nano-imaging contrast**. To systematically investigate the mechanism for the nano-imaging of buried domain wall topology through excitation of the phonon

polariton in the capping h-BN, we fabricated the h-BN encapsulated devices with small twisted angle $\theta \sim 0.1°$ with some variation due to the local strain and cracks. The bottom h-BN is 26 nm while the top h-BN is 8–16 nm. An optical microscope image (Fig. 2a) is used to identify the flake location. Note that since we are now imaging the domain walls in TBG through non-aligned h-BN layers, the top layer thickness is no longer limited to <4 nm in order to excite the SPP in graphene layer. In order to extract the tunable hyperbolic dispersive response from the h-BN encapsulated TBG heterostructure, we carry out nano-Fourier transform infrared spectroscopy (nano-FTIR) with a broadband difference frequency generation laser source at various locations, sampling the spectrum for both uniform AB/BA domain regions and domain walls regions. All the locations are identified through nano-imaging first (Fig. 2b, inset).

We collected the spectra in Fig. 2b at the blue spot location in the AB/BA domain and at the red spot location on the domain wall. Interestingly, the resonance of the phonon polariton in the h-BN/TBG heterostructure is modified by the domain wall where the resonance mode is damped and blueshifted by $\sim 5\,\mathrm{cm}^{-1}$. The underlying spectral modification can be explained by the local conductivity changes between the AB regime and the domain wall. Note that although the TBG is not specially treated with electric doping by gating, we estimate the Fermi energy to be ~50 meV due to unintentional doping of the sample by carrying out a two-probe resistance measurement at 10 K, (which is a typical value for high quality exfoliated graphene layers, see Supplementary Note 1). We then calculate the local conductivity using a tight-binding model for the two cases in Fig. 2d (Supplementary Note 2) and find significant difference at the frequency of the upper RB region of phonon polariton in h-BN. The sudden increase of the real part of the conductivity and the negative imaginary part of the conductivity make the domain wall an optically denser medium where the highly confined phonon polariton mode is dissipated and reflects back at the domain wall efficiently. This is due to the fact that the polaritons are tightly confined to the heterostructure, and the smallest perturbation is sufficient to create an efficient scattering mechanism. As shown in Fig. 2c, the theoretical prediction of the modification in the spectra using finite-difference time-domain (FDTD) simulation matches the experimental result in terms of the reduction of the amplitude (Supplementary Note 3). The frequency-dependent amplitude modification provides us not only the insight of the phonon polariton imaging mechanism, but also guidance as to how to find the optimized frequency for maximizing the nano-imaging contrast. We note that a complete detailed understanding of the frequency dependent optical conductivity at the domain wall requires further study and experimental support.

We can also extract the dispersion relation that determines the phonon polariton wavelength, which is another important parameter for the imaging resolution. To access to the wavelength for the propagating polariton wave, we scan the s-SNOM tip across the edge of the heterostructure. The field scattered by the tip is the superposition of two contributions. (i) a local contribution that depends on the sample just below the tip, and (ii) the polaritons launched by the tip and reflected back to the tip itself by the edge. The interference of these contributions gives fields with periodicity equal to half of the polariton wavelength $\lambda_p$. Figure 2e presents nano-imaging amplitude data at representative frequencies of $\omega = 1430\,\mathrm{cm}^{-1}$ and $\omega = 1530\,\mathrm{cm}^{-1}$. We observe a clear polariton fringe difference in the two cases due to the dispersion relation of the polariton modes. Line profiles are acquired perpendicular to the edge of the heterostructure allowing measurement of the polariton wavelength (Fig. 2f). The observed polariton wavelength decreased from $1160 \pm 20$ nm to $202 \pm 8$ nm when excitation frequency increased. To get the full

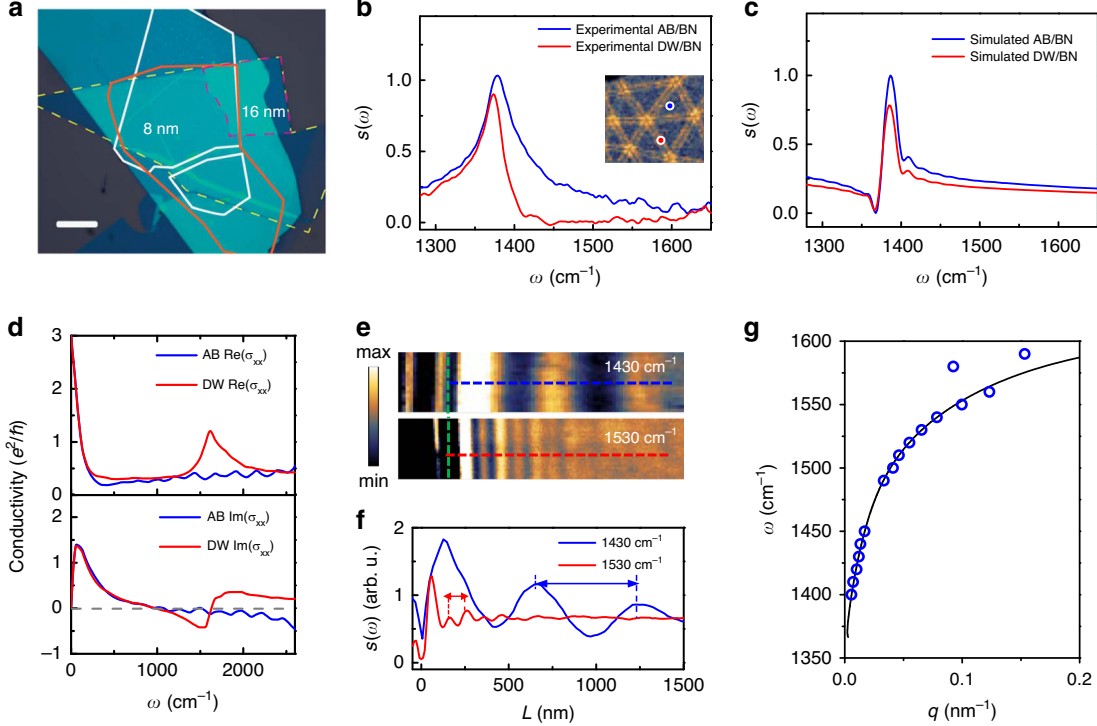

**Fig. 2 Nano-FTIR spectra and phonon polariton dispersion of encapsulated TBG heterostructure. a** Optical microscope image of the twisted bilayer graphene (red solid and dash) encapsulated with 8 nm (16 nm at folded region), top h-BN (yellow solid) and 20 nm bottom h-BN. Scale bar: 10 μm. **b** Nano-FTIR spectra taken at AB/BA domain (blue solid line) and on the domain wall (DW) (red solid line). Inset: domain walls imaged through phonon polariton in h-BN. The blue and red dots indicate the tip locations where nano-FTIR spectra were taken at AB/BA and domain wall, respectively. Data were normalized to the signal of SiO$_2$/Si substrate. **c** FDTD simulated spectra calculated for AB/BA domain (blue solid line) and along the domain wall (red solid line). **d** Tight-binding model calculated frequency dependence of the real part (top) and imaginary part (bottom) of the local conductivity $\sigma_{xx}$ for the center of the soliton regions (red solid line), and the uniform regions (blue solid line). The soliton width is 10 nm and $E_F = 50$ meV. **e** Near-field amplitude image of the h-BN/graphene/h-BN at frequency $\omega = 1430$ cm$^{-1}$ (top) and $\omega = 1530$ cm$^{-1}$ (bottom). The green dashed lines indicate the edges of the heterostructure. **f** Line profiles taken along blue and red dashed lines in **e**. Double arrows indicate $\lambda_p/2$. **g** Measured dispersion relation of the phonon polariton in the heterostructure (blue circles). Black solid line is the theoretic prediction.

dispersion relation, we repeat the measurements by varying the excitation laser frequency and extracting the wavelength (Fig. 2g). The polariton momentum $q$ can be then converted from the polariton wavelength by using equation $q = 2\pi/\lambda$. Hence, we fit the experimental data with calculated dispersion relation and find excellent agreement (Supplementary Note 3). Combining the nano-FTIR spectral result and the dispersion relation, we can determine that range from $\omega = 1500$ to $1620$ cm$^{-1}$ will be the optimal frequency range to image the moiré pattern at this thickness, where the phonon polariton has large enough amplitude contrast while the polariton wavelength remains short.

**Phonon polariton reflection at domain walls**. Without chemical or electric doping, the domain wall arrays in TBG heterostructure are readily nano-imaged through the amplitude $s(\omega)$ back-scattered shown in Fig. 3a–c. The location of the domain walls are indicated by the double-line features centered at the domain walls. The spacing of the double-lines decreases as the incident excitation frequency increases yielding better resolution for large twisted angles regions. For the large moiré domain size $a \sim 400$ nm, double-line feature of one domain wall can be clearly distinguished from the neighboring one. However, for the small domain size $a \sim 200$ nm the features are merged when imaging with excitation frequency $\omega = 1500$ cm$^{-1}$ (Fig. 3a). To resolve the smaller domains, we can increase the probing frequency to $\omega = 1580$ cm$^{-1}$ (Fig. 3b). As the polariton wavelength decreases, the distance between the double lines decreases. For larger twisting

angles, with smaller moiré domain sizes, the separation between the double lines is under the resolution limit (<80 nm) (Fig. 3c). We now test the domain size limit of the phonon polariton nano-imaging of the domain wall arrays by visualizing the moiré patterns in different twisted angle $\theta$. By increasing the twisted angle from near zero to $\theta = 0.21°$, the moiré length scales decrease from $a = 225$ nm to $a = 65$ nm (Fig. 3d–f). Given the tip radius of ~25 nm, the largest twisted angle of the TBG is about $\theta = 0.25°$ when image with frequency $\omega = 1560$ cm$^{-1}$.

The physical origin of the double-line features can further be probed by exploring the frequency dependence of the phonon polariton reflectance interference at the domain walls. Figure 4a shows phonon polariton interference profiles at three representative frequencies. In each side of the domain wall, the amplitude maxima are formed by the interference between the forward propagating wave launched by the AFM tip and the backward propagating wave reflected at the domain wall, which is a similar behavior shown in Fig. 2e where the interference happens at the edge of the heterostructure. The interference at each side of the domain wall is the same, hence forming the two symmetric peaks pattern. Obviously, the distance between the two symmetric peaks $d_{peak}$ in the three panels of Fig. 4a becomes shorter when we increase the excitation frequency, as a result of the decreasing of the phonon polariton wavelength. By varying the excitation frequencies, we plot the $d_{peak}$ as a function of the excitation frequency $\omega$ (Fig. 4b). Although the trend looks very similar to the data in Fig. 2f, the precise mechanism for the phonon polariton interference pattern at the domain walls is not exactly as

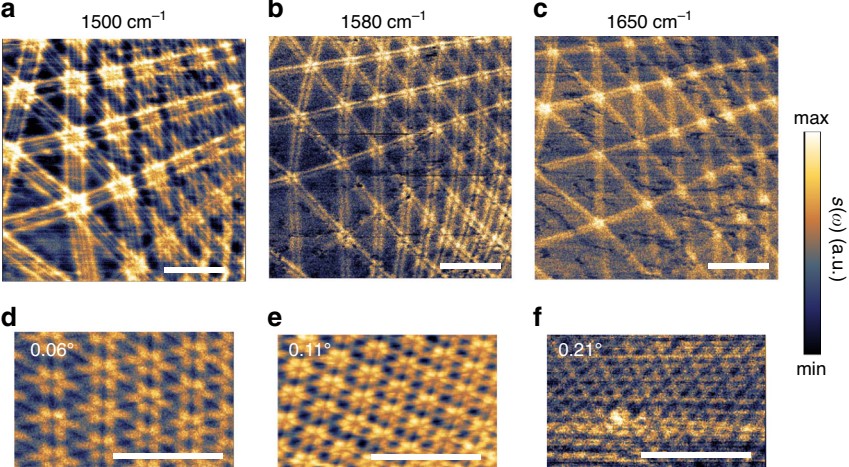

**Fig. 3 Nano-imaging of moiré pattern in buried TBG encapsulated with h-BN. a–c** Near-field images of the normalized amplitude $s(\omega)$ showing moiré patterns in TBG taken at same position with different excitation frequencies. The twisted angle is ~0.05°. **d–f** Near-field images of the normalized amplitude $s(\omega)$ showing different twisted angles of 0.06°, 0.11° and 0.21° at excitation frequency $\omega = 1560$ cm$^{-1}$. Scale bars are 500 nm.

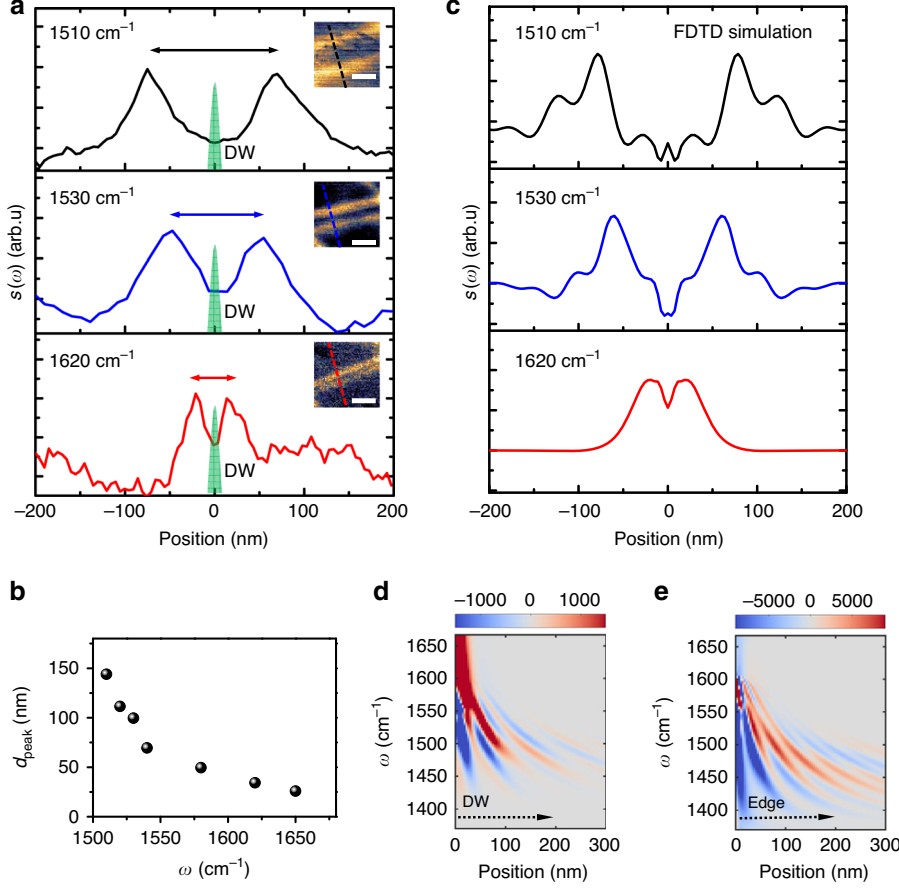

**Fig. 4 Phonon polariton reflectance at the domain walls in the underlying TBG. a** Phonon polariton interference profiles across the same domain wall (DW) under excitation frequency $\omega = 1510$ cm$^{-1}$ (top panel), $\omega = 1530$ cm$^{-1}$ (middle panel) and $\omega = 1620$ cm$^{-1}$ (bottom panel). The green shaded areas label the domain wall with width of about 10 nm. The double arrows in the top (black), middle (blue), and bottom (red) indicate the distance between the two symmetric peaks $d_{peak}$. Insets: Mapping of amplitude $s(\omega)$ of the domain wall. Dash lines indicate where the profiles are taken respectively. Scale bars: 100 nm. **b** Distance between the two symmetric peaks $d_{peak}$ (black dots) as a function of excitation frequency $\omega$. All data points are extracted at the same domain wall. **c** FDTD simulations of the phonon polariton interference profiles shown in **a**. **d, e** Dispersion of the phonon polariton interference profile calculated with FDTD simulations for frequencies $\omega = 1380$ cm$^{-1}$ to $\omega = 1670$ cm$^{-1}$ while varying positions of the sampling tip with respect to the domain wall (**d**), and the edge of the heterostructure (**e**). The simulated tip scanning directions are indicated by the dashed arrows.

same as the interference pattern at the edge of the heterostructure since the reflection phase shift plays a critical role here. It is well known that the constructive interference occurs when the phase difference between the waves is an even multiple of $\pi$[27]. Therefore, we define the effective phase shift $\varphi$ by $\varphi = 2\pi(d_{peak}/\lambda_p)$. For the interference pattern at the domain walls, we find $d_{peak} \approx \lambda_p$, indicating an effective phase shift of $2\pi$. In comparison the reflectance at the edge of the heterostructure has an effective phase shift of $\pi$ with $d_{peak} \approx \lambda_p/2$. The effective phase shift of the phonon polaritons at domain walls is indeed different than the surface plasmon polariton reflection at domain walls[16]. Therefore, we carry out FDTD numerical simulation of the polaritons reflection to confirm the validity of our interpretation (see "Methods" and Supplementary Note 4). The simulation of the polariton profile for different distances from the edge (Fig. 4c) agrees well with the experimental data (Fig. 4a) for three different choices of the excitation wavelengths and captures well the dependence of the profile on the wavelength. Figure 4d shows the profile plotted as a function of the wavelength and the position with respect to the tip, illustrating the dispersion of the reflected polaritons. Two polariton modes (the fundamental mode and the first higher order mode) contribute to the simulated fringe pattern. As a comparison, the reflection from the edge of the sample shows a $\pi$ phase shift between the two cases (Fig. 4e), which also agrees with our experimental observations. The reflection coefficient and its phase are determined by the difference of the conductivity of the domain wall with respect to the AB region (Supplementary Note 4). Combining with the simulation of the near-field IR spectrum, we unveiled the way local conductivity affects both the phonon polariton amplitude and reflection phase change at domain walls providing much insight onto the optical properties of domain wall solitons in TBG.

## Discussion

In conclusion, we show that phonon polaritons launched in h-BN capping layers exhibit reflection at domain wall arrays generated in buried heterostructures below and can be used to directly map and visualize the moiré superstructure in the vdW heterostructure using near-field IR microscopy. This nondestructive method will be particularly useful for control of the fabrication processes where the atomic stacking orders of vdW heterostructures can be unintentionally changed due to temperature or strain variations. Furthermore, since the top and bottom h-BN layers are utilized as the medium to launch the tunable phonon polariton modes, this method is a nondestructive, in situ approach. In addition, the possible layer thicknesses of the protective h-BN are much more flexible compared to other methods which only allow few nm top h-BN layer preventing high quality electrostatic gating. Our result paves a path to in situ testing in these complex systems utilizing phonon polaritons to observe changes as we optimize our ability to manipulate and control the local conductivity in vdW heterostructures.

## Methods

**Sample preparation**. Samples were fabricated by dry transfer of mechanically exfoliated flakes. Polycarbonate film was used as the transfer polymer and the graphene bilayer was formed by tearing and stacking a single monolayer with an imposed twist angle in the range of 0.1–0.3°[10]. Top BN ranging from 4 to 16 nm thick and bottom BN of around 20 nm were used to encapsulate the TBG. Samples were not annealed or heated above 180 °C to reduce the chance of twisting back to 0°. Note for direct plasmon to phonon-polariton response comparison, samples with doping induced by oxygen plasma treatment were also prepared.

**Near-field optical measurement**. The nano-imaging and nano-FTIR experiments were performed using s-SNOM (Neaspec GmbH). The system is equipped with continuous wave mid-IR quantum cascade lasers (DRS Daylight Solutions Inc.) for

nano-imaging and broad-band difference frequency generation lasers for nano-FTIR. The platinum silicon coated tip used in the s-SNOM has a typical radius of 20 nm operating in the tapping mode with a tapping frequency around 300 kHz. We use both pseudo-heterodyne interferometric detection and non-interferometric detection module to extract the near-field signal. The background signal is suppressed by demodulation of the near-field signal at the third harmonics of the tapping frequency. All the near-field optical measurements are done at room temperature.

**FDTD simulation**. FDTD simulations are performed in Lumerical FDTD Solutions. h-BN is modeled with a Lorentz model while TBG is modeled with 2D rectangles (boundary conditions). Away from the domain walls, we used conductivity that is computed for the AB regime in the TBG. For the simulation of the profile of polaritons reflected at the domain walls and the heterostructure edge, the mesh size was set to 1 nm, and the tip was emulated with a vertically polarized Hertzian dipole placed at the interface between the top h-BN and air combined with electric field monitors at different heights just above the Hertzian dipole. The background can be simulated at a location far from any discontinuity and subtracted frequency-by-frequency from the map to remove the background (see Supplementary Notes for more details about the post processing).

## Data availability

The data that support the plots within this paper and other findings of this study are available from the corresponding author upon reasonable request.

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

## Acknowledgements

Part of the work was performed at Center for Nanoscale Systems at Harvard University support by the National Science Foundation (NSF) under award NNCI-1541959. Y.L. was supported by the Department of Energy under award DE-SC0019300. P.K. acknowledges support from DoD Vannevar Bush Faculty Fellowship (N00014-18-1-2877). M.M. and E.K. acknowledge support from ARO MURI award No. W911NF-14-1-0247.

## Author contributions

Y.L., E.K., P.K., and W.L.W. conceived the experiment. R.E. fabricated heterostructure samples. Y.L. and R.E. carried out the experiments and analyzed the data. M.M. and S.C. performed theoretical calculations. M.T. and Y.L. performed FDTD simulations. All authors discussed results and wrote the manuscript.

## Competing interests

The authors declare no competing interests.
