## [Peer Review File · Nature Communications]

REVIEWER COMMENTS

Reviewer #1 (Remarks to the Author):

In this manuscript, Luo et al. reported the characterization of h-BN encapsulated twisted bilayer graphene heterostructure by s-SNOM. Through the measurement of the buried heterostructure, the hyperbolic h-BN phonon polaritons launched in the h-BN layer interacting with the moiré structure in the twisted bilayer graphene was explored. By comparing experimental results and theoretical calculations, the influence of domain walls and the formation mechanism of the hyperbolic phonon polariton are interpreted. Their results are novel and should attract interest from many researchers in the fields of 2D materials. However, there are several points that need to be clarified before this work is considered for publication.

1. Line 108, what is the meaning of "the type II region". Where is "the type I"? What is their difference.
2. In Fig 3a, there are two symmetric peaks formed at the both sides of domain walls. Does this mean that the domain walls have inherent width? If the width of the domain walls cannot be ignored, its influence on the value of d_{peak} here should be considered.
3. As shown in Fig 4c, there are second-order peaks observed in the calculation results. Why the second-order peaks are not seen in the experimental results. Pls explain.
4. Line 119, "Note that since we are now imaging the domain walls in TBG through nonaligned h-BN layers, the top layer thickness is no longer limited to less than 4nm in order to excite the SPP in graphene layer." May I know the reason why the top layer thickness should be less than 4nm if the graphene/hBN has a very small separation angle ([Nature Communications 6, 6499, 2015.] [Nature Materials 14, 1217–1222, 2015.]).
5. Line 29, there are two "in" in the sentence.
6. Information of scale bars in Fig 1c-e, 2a is missing.
7. In Fig 3b, it shows the laser frequency of 1580cm⁻¹, while it mentioned in the manuscript is 1530cm⁻¹ (line 171). Pls double check.

Reviewer #2 (Remarks to the Author):

The article "In-situ nanoscale imaging of moiré superlattices in twisted van der Waals heterostructures" by Y. Luo et al. demonstrates that the moiré pattern in twisted by-layer graphene can be indirectly imaged by imaging the phonon polariton interference on the top hBN encapsulation layer. Due to optical conductivity difference in the domain wall, the measured polariton interference image is reminiscent to the underlying moiré pattern. This nondestructive imaging method shows several advantages over other commonly used imaging techniques such as STM, TEM, and PFM.

The authors provided sufficient experimental evidence as well as simulation and modelling efforts. I don't see major technical issues. However, the resolution of this imaging technique seems to be limited by the tip radius and phonon polariton wavelength to ~80 nm, which means samples with larger twist angle is not applicable. Currently, the more interesting samples typically have larger twist angle, namely ~1.1 degree for obvious reasons. Therefore, I foresee the usefulness of the proposed imaging technique to be very limited. Besides, I list several more detailed issues below:

1. The length of the scale bars in Figure 1 c, d, and e are missing, therefore it's unclear how large the moiré pattern is. The twist angle is also not mentioned.
2. In Figure 2 c the calculated spectra are maximum in the $|r_p|^2$. The authors claim this qualitatively reflects the near-field signal. I wonder if any previous work or theory can support this.
3. The simulated spectra in Figure 2 c are obtained by simulating a pyramid shape tip above the sample. There are several problems.
 - First of all, the authors did not specify the tip parameters like length, apex radius, opening angle,

etc. Boundary conditions, sample size etc are not mentioned as well.

- Secondly, the power absorbed by the sample is calculated by the power field monitor between the tip and sample. Why does it reflect the near-field scattering?
- What's the tip-sample distance?
- How is the oblique incident angle handled in the simulation? When calculating a broadband oblique incidence, the incident angle is usually not uniform using a FDTD solver.

4. In line 154, the sentence reads "Line profiles are acquired perpendicular to the edge of the heterostructure allow measurement of the polariton wavelength (Fig. 2f)." There is some grammar issue with this sentence.

In general, I believe this manuscript by Luo et al. presents a nice work on electronic moiré patterns in graphene using near-field mapping of the hBN polaritons. The results are solid but no new physics insights have been provided. I recommend major revision.

Reviewer #3 (Remarks to the Author):

In the manuscript "In-situ nanoscale imaging of moiré superlattices in twisted van der Waals heterostructures," Y. Luo et al., have addressed the reflection of phonon-polaritons in hBN off of a buried soliton lattice in TBG. Overall, I believe the topic is interesting and important. However, the manuscript lacks clarity and sufficient detail to fully understand the advances claimed within this work. I believe that significant revisions are needed before the work is ready for publication.

1. Emphasis is placed on the utility of non-destructive imaging of nano-meter scale properties in moiré superlattices in van der Waals (vdW) heterostructures. Yet the authors technique is limited to extremely small twist angles $<0.250^\circ$, which is well below the magic angle, and limits the potential utility as a characterization tool. Do the authors have a suggestion for how the technique can be extended to larger twist angles?

2. One of the main advantages emphasized by the authors appears to be the ability to interrogate moiré patterns in TBG through slightly thicker (8-16 vs. 4 nm) hBN capping layers. It is claimed several times that the domain walls cannot be imaged through hBN top layers thicker than 4 nm with the method demonstrated by S. Sunku et al. Yet in the authors own Fig 1d the domain walls from the AB/BA domains can be observed at 940 cm^{-1} , which is outside of the reststrahlen band of hBN. Are the measurements presented in Fig. 1d and e performed on the same device with the same doping level? The authors should clarify or modify their claims.

3. The authors note that the previous work of Sunku et al., has the disadvantage that the graphene layer must be sufficiently doped (lines 56 to 60). In the present work the graphene is also doped ($\sim 50\text{ meV}$). Are the authors claiming that the doping of their device is unimportant? Can the authors clarify how doping impacts the nano-infrared contrast?

4. The authors invoke a $\sim 0.2\text{ eV}$ resonance in the optical conductivity to explain the nano-infrared spectroscopic results at the domain wall. The authors should state their procedure for normalizing the data presented in Fig. 2b as it is essential to the conclusions. It is also not so obvious what is the influence of the propagating polariton mode on the spectra recorded near the 'DW location'. A hyperspectral line scan across the domain wall may be more appropriate for this situation.

5. The frequency dependent optical conductivity is used to model the nano-optical spectra with a rather un-conventional method of comparing the absolute value of R_p at the first order maxima of the hyperbolic polariton in hBN to, qualitatively, explain the experimental data of the near-field amplitude. Why did the authors not use the lightning rod or even point dipole model? As is, a complete explanation as to what the precise meaning of, and we can learn from, the R_p comparison is merited.

6. The authors also use FDTD, described in Supplementary Note 4. The latter section describes in great detail what was simulated. However, it does not clarify if the simulations of z-component of the electric field were put through a demodulation scheme to arrive at the trend displayed in Fig. 2c and does not clearly state what is plotted. Thus, it would help to clarify what exactly is plotted in Fig 2c and provide an appropriate label of the vertical axis. The end result is claimed to qualitatively resemble the experimental data reproducing the observation that 'the resonance mode is damped and blue shifted.' Which appears dodgy as enhancement of the near-field amplitude is maximal at whatever frequency β [Ref-McLeod PRB 90, 085136 (2014)] is maximal and an apparent 'shift' in the amplitude spectra could be caused by a dispersion-less offset of the optical conductivity. The spectra may be further complicated by the propagating polariton as in comment 4. Thus, I have several questions. Is the calculated optical conductivity capable of explaining the DW contrast observed at 940 cm^{-1} in Fig 1d? Is a resonance in the optical conductivity at 0.2 eV essential to explain the experimental findings, or is the motivation for the latter resonance purely theoretical? Is the predicted resonant frequency dependent on the twist angle of the TBG?

7. The authors may wish to address momentum in the introduction of launching polaritons with s-SNOM (lines 75-78). As written the text could be interpreted as polaritons are launched because of the enhanced electric field. "The appropriate mid IR frequency" is, also, not clearly defined.

8. The sentence "This advance offers the ability to image the physics of pre and post processed devices undertest" (line 86) appears dodgy. What does the statement "image the physics" mean precisely?

9. Several other phrases are somewhat obscure:

a. Line 25 'spatially dependent field profile' – what field?

b. Line 85 'quasi-particle dynamics' – what dynamics are the authors referring to? The utility of dynamic hybridization has furthermore been pointed out by several authors, including buried surfaces [see for example, K. Chaundhary et al., Science Advances 5, eaau7171 (2019)].

c. Line 139 'the smallest perturbation is sufficient to create an efficient scattering mechanism.' The meaning of 'the smallest perturbation' is unclear, which relates to comment 2.

d. Line 221 'electronic topology' appears very suddenly and its relevance to the discussion is not clarified within the text.

Reviewer #1

General comment: *In this manuscript, Luo et al. reported the characterization of h-BN encapsulated twisted bilayer graphene heterostructure by s-SNOM. Through the measurement of the buried heterostructure, the hyperbolic h-BN phonon polaritons launched in the h-BN layer interacting with the moiré structure in the twisted bilayer graphene was explored. By comparing experimental results and theoretical calculations, the influence of domain walls and the formation mechanism of the hyperbolic phonon polariton are interpreted. Their results are novel and should attract interest from many researchers in the fields of 2D materials. However, there are several points that need to be clarified before this work is considered for publication.*

Our response: First, we like thank the reviewer for pointing out the novelty of the work and the interest and impact it will have on the community.

Comment 1: *Line 108, what is the meaning of “the type II region”. Where is “the type I”? What is their difference?*

Our response: Here the type I and type II regions refer to the two hyperbolic regions of h-BN, which are also called reststrahlen bands (RB), this nomenclature was introduced by Dai et al (Nat. Nanotechnol, 10,682 (2015)) . For clarity, we will change the notation to RB1 for the lower reststrahlen band and RB2 for the upper reststrahlen band in the revised manuscript. The RB1 band ranges from 746 to 819 cm^{-1} , where the out-of-plane h-BN permittivity ϵ_z is negative. The RB2 band ranges from 1370 to 1610 cm^{-1} , where the in-plane h-BN permittivity $\epsilon_{x,y}$ is negative. We focus on the RB2 band in this work due to the strong interaction between the TBG and the h-BN polariton mode.

Comment 2: *In Fig 3a, there are two symmetric peaks formed at the both sides of domain walls. Does this mean that the domain walls have inherent width? If the width of the domain walls cannot be ignored, its influence on the value of d_{peak} here should be considered.*

Our response: The domain walls have a fixed width of ~8 nm for the small twisted angles (Alden et al. PNAS, 110,11256 (2013)). This width is much smaller than the d_{peak} (50-150nm) we observed. The measured value is a complex convolution with the tip geometry. Given this, the width of the domain walls can be reasonably ignored. In fact, the two symmetric peaks formed at the both sides of domain walls are primarily driven by polariton reflection, as we discuss in the latter parts of the manuscript.

Comment 3: *As shown in Fig 4c, there are second-order peaks observed in the calculation results. Why the second-order peaks are not seen in the experimental results. Pls explain.*

Our response: The FDTD simulation is very sensitive and can reveal higher order detail. These second order peaks are due to higher order propagating modes which are captured by the simulation. It is important to note that higher order polaritons are weaker, occur at much larger momenta and are very sensitive to the tip/experimental geometry. Therefore, in the “real” experiment, where AFM tips a not “perfect” sharp probes and where samples have finite surface roughness, experimental SNR makes the detection of higher order signals extremely difficult. Higher order phonon polariton modes have been observed only under very controlled conditions where the sample has to be thick and suspended from the substrate (Fali et al. Nano Lett, 19, 7725 (2019)). In our case, the h-BN is only ~30 nm in total (top and

bottom) and cannot be suspended from the substrate. We added this note in the revised Supplementary Information.

Comment 4: Line 119, “Note that since we are now imaging the domain walls in TBG through nonaligned h-BN layers, the top layer thickness is no longer limited to less than 4nm in order to excite the SPP in graphene layer.” May I know the reason why the top layer thickness should be less than 4nm if the graphene/hBN has a very small separation angle ([Nature Communications 6, 6499, 2015.] [Nature Materials 14, 1217–1222, 2015.]).

Our response: In previous work, thin h-BN layers were needed because of direct *plasmon polariton* excitation of TBG through h-BN was desired. In both works mentioned by the reviewer, the graphene layers were directly grown on top of h-BN, thereby no additional layer was in between graphene and the SNOM tip. In contrast, we excite the h-BN *phonon polariton* and take advantage of the interfacial dynamics to reveal the moiré geometry. Therefore, we are not limited to the thin h-BN top layer.

Comment 5: Line 29, there are two “in” in the sentence.

Our response: We thank the reviewer for pointing this out. We deleted the extra “in” in the sentence.

Comment 6: Information of scale bars in Fig 1c-e, 2a is missing.

Our response: We thank the reviewer for pointing this out. We added the missing information of the scale bars.

Comment 7: In Fig 3b, it shows the laser frequency of 1580cm⁻¹, while it mentioned in the manuscript is 1530cm⁻¹ (line 171). Pls double check.

Our response: We thank the reviewer for pointing this out. We corrected the laser frequency to be 1580 cm⁻¹.

Reviewer #2

General comment: *The article “In-situ nanoscale imaging of moiré superlattices in twisted van der Waals heterostructures” by Y. Luo et al. demonstrates that the moire pattern in twisted by-layer graphene can be indirectly imaged by imaging the phonon polariton interference on the top hBN encapsulation layer. Due to optical conductivity difference in the domain wall, the measured polariton interference image is reminiscent to the underlying moire pattern. This nondestructive imaging method shows several advantages over other commonly used imaging techniques such as STM, TEM, and PFM. In general, I believe this manuscript by Luo et al. presents a nice work on electronic moire patterns in graphene using near-field mapping of the hBN polaritons. The results are solid but no new physics insights have been provided. I recommend major revision.*

Our response: We thank the reviewer for recognizing our manuscript presents a “*nice work on electronic moire patterns in graphene using near-field mapping of the hBN polaritons, with solid results.*” We disagree with the notion that the work provides no new physics insights. The ability to probe moire structure and moire physics “*in-situ*”, *non-destructively*, gives researchers the ability to image devices under test or during processing, an important need.

Major comment: *The authors provided sufficient experimental evidence as well as simulation and modelling efforts. I don’t see major technical issues. However, the resolution of this imaging technique seems to be limited by the tip radius and phonon polariton wavelength to ~80 nm, which means samples with larger twist angle is not applicable. Currently, the more interesting samples typically have larger twist angle, namely ~1.1 degree for obvious reasons. Therefore, I foresee the usefulness of the proposed imaging technique to be very limited.*

Our response: The reviewer correctly pointed out that the spatial resolution of our technique is ~ 50 nm due to the limitation of tip radius and phonon-polariton wave length excited by the light we used in this experiment and the convolutional nature of the technique. While the spatial resolution can in principle be improved by using sharper tips and smaller wavelength down to 20 nm, the length scale is still larger than the moire length scale of a magic angle twisted bilayer, ~ 10 nm. We, however, wish to point out that there are still numerous exciting problems that our experimental techniques can be directly applicable. Several examples include: (1) Moire domain reconstruction [Yoo et al., Nature Mat 2019]; (2) Topological transport in network of domain boundaries [Rickhaus, P. et al. Nano Lett. 18, 6725 (2018)]; and (3) Mott insulator and topological excitons in twisted bilayer TMDs (Andersen et al., arXiv:1912.06955). These are only a few small number of examples. In addition, the non-destructive nature of the technique could be an important value.

Comment 1: *The length of the scale bars in Figure 1c, 1d, and 1e are missing, therefore it’s unclear how large the moire pattern is. The twist angle is also not mentioned.*

Our response: We thank the reviewer for pointing this out. We added the missing information of the scale bars for clarity. We also added the twist angle in the figure caption.

Comment 2: *In Figure 2 c the calculated spectra are maximum in the $|r_p|^2$. The authors claim this qualitatively reflects the near-field signal. I wonder if any previous work or theory can support this.*

Our response: The reflection coefficient r_p is a fundamental response function of the system that describes the relative magnitude and phase of p -polarized light reflected from the surface of the material with a frequency-dependent dielectric function $\epsilon(\omega)$ [PRB 85, 075419 (2012); PRB 90, 085136 (2014)]. In the SI of PRL 117, 086801 (2016), the total reflection $|r_p|$ has been used for a rough estimation of the scattered near-field. We have added a citation of PRL 117, 086801 (2016) in the last paragraph of our Supplementary Note 3 for clarity.

Comment 3: *The simulated spectra in Figure 2c are obtained by simulating a pyramid shape tip above the sample. There are several problems. First of all, the authors did not specify the tip parameters like length, apex radius, opening angle, etc. Boundary conditions, sample size etc are not mentioned as well. Secondly, the power absorbed by the sample is calculated by the power field monitor between the tip and sample. Why does it reflect the near-field scattering? What's the tip-sample distance? How is the oblique incident angle handled in the simulation? When calculating a broadband oblique incidence, the incident angle is usually not uniform using a FDTD solver.*

Our response: We used a simulation region of $4\mu\text{m} \times 4\mu\text{m}$, and the boundary conditions are PML (perfectly matched layer). The tip is approximated with a square truncated pyramid with base of 20 nm, opening angle of 12 degrees, height 2.4 microns. When placing the monitor extremely close to the sample surface (few nanometers), the reflected field can be used to approximate the scattered field similar to the dipole radiation pattern. To verify that, we carried out the simulation with the monitor $2\mu\text{m}$ away from the tip and perpendicular to the sample surface to measure the scattered field. As shown in Fig.1, the result is very close to data in Fig.2c of the manuscript. Our tip-sample distance is 40 nm. The incidence angle is 60° as same as the experimental setup. To handle this, we used a special plane wave source called Broadband Fixed Angle Source Technique (BFAST) which gives special treatment to the broadband oblique incidence light source in combination with PML boundary condition. We added these details to the revised Supporting Information.

Fig.1 Near-field amplitude signal $s(\omega)$ simulated with a monitor far away from the tip.

Comment 4: *In line 154, the sentence reads “Line profiles are acquired perpendicular to the edge of the heterostructure allow measurement of the polariton wavelength (Fig. 2f).” There is some grammar issue with this sentence.*

Our response: We thank the reviewer for pointing this out. We corrected this sentence to read “Line profiles are acquired perpendicular to the edge of the heterostructure allowing direct measurement of the polariton wavelength (Fig. 2f).”

Reviewer #3

General comment: *In the manuscript “In-situ nanoscale imaging of moiré superlattices in twisted van der Waals heterostructures,” Y. Luo et al., have addressed the reflection of phonon-polaritons in hBN off of a buried soliton lattice in TBG. Overall, I believe the topic is interesting and important. However, the manuscript lacks clarity and sufficient detail to fully understand the advances claimed within this work. I believe that significant revisions are needed before the work is ready for publication.*

Our response: We thank the reviewer to point out that our manuscript is interesting and important. We revised the manuscript to improve clarity and expand details, based on their recommendations.

Comment 1: *Emphasis is placed on the utility of non-destructive imaging of nano-meter scale properties in moiré superlattices in van der Waals (vdW) heterostructures. Yet the authors technique is limited to extremely small twist angles <0.250 , which is well below the magic angle, and limits the potential utility as a characterization tool. Do the authors have a suggestion for how the technique can be extended to larger twist angles?*

Our response: The reviewer correctly points out that the spatial resolution of our technique is currently ~ 50 nm due to the limitation of tip radius and phonon-polariton wave length excited by the light we used in this experiment. While the spatial resolution can in principle be improved by using sharper tips and smaller wavelength, down to 20 nm, the length scale is still larger than the moire length scale of the magic angle twisted bilayer, ~ 10 nm. We, however, wish to point out that there are still numerous exciting problems that our experimental techniques can be directly applicable. Several examples include: (1) Moire domain reconstruction (Yoo et al., Nat. Mat 2019); (2) Topological transport in network of domain boundaries (Rickhaus et al. Nano Lett. 18, 6725 (2018)); and (3) Mott insulator and topological excitons in twisted bilayer TMDs (Andersen et al., arXiv:1912.06955). These are only a few small number of examples. In addition, the non-destructive nature of the technique could be an important value.

Comment 2: *One of the main advantages emphasized by the authors appears to be the ability to interrogate moiré patterns in TBG through slightly thicker (8-16 vs. 4 nm) hBN capping layers. It is claimed several times that the domain walls cannot be imaged through hBN top layers thicker than 4 nm with the method demonstrated by S. Sunku et al. Yet in the authors own Fig 1d the domain walls from the AB/BA domains can be observed at 940 cm^{-1} , which is outside of the reststrahlen band of hBN. Are the measurements presented in Fig. 1d and e performed on the same device with the same doping level? The authors should clarify or modify their claims.*

Our response: We thank the reviewer for pointing the lack of clarity here. The measurements presented in Fig 1d and e *are performed* on the same device with the same doping level. The device substrate was treated with oxygen plasma to increase the carrier density. Similar to the sample used by S. Sunku et al, we used 4nm h-BN capping layer in this case. Fig.1d is measured at 940 cm^{-1} where we directly excite and detect the graphene plasmon polariton (Line 102). In contrast, we measured at 1420 cm^{-1} for Fig. 1e to excite and detect the h-BN phonon polariton. They both reveal the same moiré patterns in the TBG (Line 107-108).

Comment 3: *The authors note that the previous work of Sunku et al., has the disadvantage that the graphene layer must be sufficiently doped (lines 56 to 60). In the present work the graphene is also doped*

(~50 meV). Are the authors claiming that the doping of their device is unimportant? Can the authors clarify how doping impacts the nano-infrared contrast?

Our response: The device shown in Fig.2a is *naturally* doped with $E_F \sim 50$ meV, comparing to the sample with $E_F \sim 300$ meV in the work by Sunku et al. As shown in Fig.1 (adapted from Ref. 18), the graphene plasmon can only be lossless when E_F is around 300 meV. For lower Fermi level ($E_F = 50$ meV), the plasmons are either too confined or they suffer high losses due to interband transitions dominate, thereby hard to detect any plasmon polariton with a reasonable propagation constant. However, in our case, we use phonon polariton in *h*-BN to image the moiré patterns instead of using graphene plasmon. The nano-infrared contrast is determined by the local conductivity change at the domain walls, which is fairly independent to the doping level of the graphene layer.

Fig.2 Graphene plasmon polariton dispersion under different doping levels.

Comment 4: The authors invoke a ~0.2 eV resonance in the optical conductivity to explain the nano-infrared spectroscopic results at the domain wall. The authors should state their procedure for normalizing the data presented in Fig. 2b as it is essential to the conclusions. It is also not so obvious what is the influence of the propagating polariton mode on the spectra recorded near the 'DW location'. A hyperspectral line scan across the domain wall may be more appropriate for this situation.

Our response: To normalize the data shown in Fig. 2b, we first measure the IR signal from the SiO₂/Si substrate $s_{\text{ref}}(\omega)$. With the same measurement condition, we then move the tip to AB/BA regime and domain walls on the TBG stack to acquire the signal and normalize to $s_{\text{ref}}(\omega)$. All data are relative value with arbitrary unit. We added this clarification in the figure caption. We agree with the reviewer that a hyperspectral line scan across the domain wall will be helpful. However, the sample stage of the nano-FTIR setup has non-negligible drift over a long (time-scale) scanning process. Unlike the nano-imaging, spectrum capture has weak signal and requires much longer integration time at each tip position. Therefore, an accurate continuous multipoint line scan takes a long time and is therefore extremely challenging. Instead, we take multiple measurements at the AB/BA regime and near the DW location respectively. In this way we avoid the sample stage drifting by manually relocate at the DW position. We believe this is a more accurate alternative to a hyperspectral line scan, given current experimental conditions. Furthermore, in our FDTD simulation (Fig.4d in the manuscript), we performed line-scans across the domain wall, which is in a good agreement to our experiment result.

Comment 5: The frequency dependent optical conductivity is used to model the nano-optical spectra with a rather un-conventional method of comparing the absolute value of R_p at the first order maxima of the hyperbolic polariton in *h*BN to, qualitatively, explain the experimental data of the near-field amplitude. Why did the authors not use the lightning rod or even point dipole model? As is, a complete explanation as to what the precise meaning of, and we can learn from, the R_p comparison is merited.

Our response: A straightforward approximation for understanding the s-SNOM signal is given by the reflection of the polariton (r_p) approximation that captures the overall near field behavior. Similar to the previous studies (Dai et al. Science 343, 1125 (2014) and Dai et al. Nat. Nanotechnol 10, 682 (2015)), we use the r_p approximation to understand the underlying physics of our near field experiment, since r_p approximation is the simplest model, and its equations are analytical. However, this model introduces a blue shift because it ignores tip-sample coupling (PRB 85, 075419 (2012) and PRB 90, 085136 (2014)). We get a more complete picture by performing simulations that include the tip-sample interactions. We solve the time dependent Maxwell equations where the tip is considered as Hertzian dipole. The FDTD simulations are in great agreement with the experimental near field results and through the r_p model. This suggests that we sufficiently understand the physical process, therefore, we believe that there is not a substantial benefit to be gained through the use other models.

Comment 6: The authors also use FDTD, described in Supplementary Note 4. The latter section describes in great detail what was simulated. However, it does not clarify if the simulations of z-component of the electric field were put through a demodulation scheme to arrive at the trend displayed in Fig. 2c and does not clearly state what is plotted. Thus, it would help to clarify what exactly is plotted in Fig 2c and provide an appropriate label of the vertical axis. The end result is claimed to qualitatively resemble the experimental data reproducing the observation that ‘the resonance mode is damped and blue shifted.’ Which appears dodgy as enhancement of the near-field amplitude is maximal at whatever frequency β [Ref-McLeod PRB 90, 085136 (2014)] is maximal and an apparent ‘shift’ in the amplitude spectra could be caused by a dispersion-less offset of the optical conductivity. The spectra may be further complicated by the propagating polariton as in comment 4. Thus, I have several questions. Is the calculated optical conductivity capable of explaining the DW contrast observed at 940 cm^{-1} in Fig 1d? Is a resonance in the optical conductivity at 0.2 eV essential to explain the experimental findings, or is the motivation for the latter resonance purely theoretical? Is the predicted resonant frequency dependent on the twist angle of the TBG?

Our response: The reviewer is correct that demodulation scheme is commonly used in the experiment and simulation. It will effectively reduce the far-field background. However, in the FDTD simulation, we have the monitor very close to the tip and sample and simulate the near-field effect without far-field scattering. Therefore the simulated spectrum is almost same regardless of the position where it is measured. As shown in Fig.3, we performed additional simulations using the demodulation scheme to reconstruct the optical harmonics of the system. The results are very similar to the Fig.2c of the manuscript validating our assumption, especially at the resonance. When sampled right at the tip position in the simulation setup a demodulation scheme will not provide much benefit. We added this note to the Supplementary Information. In Fig.2c, we plotted the near-field signal amplitude $s(\omega)$ which is proportional to the absolute value of the electric field amplitude. We now added the missing y-axis label. For the blue shift we observed, it is different from McLeod et al explored in their work. We are measuring same thickness of graphene/h-BN heterostructure where no topographic change happens at the AB/BA region or domain walls. The optical conductivity change is not trivial but due to

Fig.3 Near-field amplitude signal $s(\omega)$ simulated with a z-position demodulation scheme.

the conductivity change between the AB/BA region and the domain wall, which is not contradict to what the reviewer interpreted. Our calculated optical conductivity is dedicated for the doping level of $E_F=50\text{meV}$ while for Fig 1d, we doped the graphene to $E_F\approx 300\text{meV}$. At $E_F=50\text{meV}$ graphene plasmon is not supported, therefore our calculation cannot explain Fig.1d. However, for that case, it has been already explained by Sunku et al (Ref. 16). It is not our goal to explore the plasmon polariton in our manuscript. The resonance at 0.2 eV alone is not essential to explain our findings. To fully explain the experimental finding, we showed in Fig.4 where we combined the optical conductivity contrast and FDTD simulation to reproduced moiré pattern imaging. Note that although we cannot use the optical conductivity we calculate in the manuscript to explain the contrast in Fig.1d, our FDTD simulation setup can still work by assuming the doping level of graphene $E_F\approx 300\text{meV}$ and a Drude model with a conductivity difference of 50% for the DW and AB regions. As shown in Fig.4, the contrast at $\omega=940\text{cm}^{-1}$ is visible as the red arrow pointed out. Such simulation indeed shows the power of FDTD simulation on this type of problems. When the twist angle is much smaller than the magic angle (1.1°) the optical conductivity of the different twisted angles does not change much. Therefore we predict that the optical contrast will not change due to the twisted angle in our interest regime, which is clearly shown in Fig.3.

Fig.4 Near-field amplitude signal $s(\omega)$ simulated across the domain wall (position 0) at lower frequency $\omega=650\text{-}1100\text{ cm}^{-1}$.

Comment 7: The authors may wish to address momentum in the introduction of launching polaritons with s-SNOM (lines 75-78). As written the text could be interpreted as polaritons are launched because of the enhanced electric field. “The appropriate mid IR frequency” is, also, not clearly defined.

Our response: The electric field is enhanced in a very small region directly below the tip. Because of the uncertainty principle, the momentum of the photons that can be coupled in the material is very large, thus allowing launching the polaritons. Without the tip, the photon momentum alone would not be sufficient to couple with the polaritons, the tip allows overcoming the momentum mismatch, and resonant excitation maximized the field coupling to the mode. We added the specific frequency range ($900\text{-}1650\text{ cm}^{-1}$) after “the appropriate mid IR frequency”.

Comment 8: The sentence “This advance offers the ability to image the physics of pre and post processed devices undertest” (line 86) appears dodgy. What does the statement “image the physics” mean precisely?

Our response: We refer here to the possibility of acquiring useful information about the local conductivity change at the domain walls of TBG via nano-imaging, which can help to elucidate the electronic landscape, as proven in this work. We changed the sentence in the revised manuscript as “This advance offers the ability to image the physics of pre and post processed devices undertest.”

Comment 9: Several other phrases are somewhat obscure:

a. Line 25 ‘spatially dependent field profile’ – what field?

b. Line 85 ‘quasi-particle dynamics’ – what dynamics are the authors referring to? The utility of dynamic hybridization has furthermore been pointed out by several authors, including buried surfaces [see for example, K. Chaundhary et al., Science Advances 5, eaau7171 (2019)].

c. Line 139 ‘the smallest perturbation is sufficient to create an efficient scattering mechanism.’ The meaning of ‘the smallest perturbation’ is unclear, which relates to comment 2.

d. Line 221 ‘electronic topology’ appears very suddenly and its relevance to the discussion is not clarified within the text.

Our response:

a. We added electric field in Line 25. Only electric field is relevant here for s-SNOM measurement.

b. “Quasi-particles dynamic” refers to the dynamics of polaritons that are formed by strong coupling between photon and phonon in h-BN or between photon and plasmon in graphene, which can be understood in a quantum mechanical theory of these collective excitations.

c. As observed by Fei et al. (Nat. Nanotechnol 8, 821(2013)) even something as narrow as a domain boundary is sufficient to create a sizeable discontinuity for plasmons and polaritons propagation in general. This is a distinctive feature of polaritons, and it is due to the fact that they exist in a quasi-static electric field configuration, meaning that the energy is stored either as electric field outside the material or inside the material in the dipole excitation of the polariton (while the contribution of the magnetic field is negligible). This means that any change of optical conductivity in the material, even if small or narrow, can launch the polaritons effectively.

d. This is a general reference to the fact that our method and SNOM in general can be used to probe exotic electron states and new materials. However, we delete it from the revised manuscript for clarity.

REVIEWER COMMENTS

Reviewer #1 (Remarks to the Author):

The authors successfully addressed my concerns. I would like to suggest to publish the manuscript.

Reviewer #2 (Remarks to the Author):

The authors have successfully addressed all my concerns and comments. I can now recommend it for publication.

Reviewer #3 (Remarks to the Author):

The authors responses are not completely satisfactory. I do not see any issues with most of their findings, specifically the very nice observation of hBN polaritons reflecting off of the domain-walls in TBG and the assignment that the latter reflection is caused by the spatially inhomogeneous optical conductivity at the domain wall. I emphasize, again, that I believe the latter observation is both interesting and important for the community. I, however, have remaining concerns with the presentation of the key scientific messages and feel that significant revisions are still needed. The two main issues I have are detailed below:

1. The authors have stated that the morie pattern has been imaged in a fully encapsulated sample with minimal doping without the aid of polaritons in the superstrate within their own work, see Fig 1d. So, the statement that <4 nm hBN superstrates and highly doped samples are needed to visualize the morie in encapsulated devices without the hBN polaritons appears to be disproved in the same report where it is claimed. If I have accurately assessed the situation, I believe these claims should be modified.

2. My major technical concern is: it is still not clear if the specific form of the frequency dependent optical conductivity reported in Fig. 2d is fully supported by the data in this report. In the rebuttal the authors have stated: "The resonance at 0.2 eV alone is not essential to explain our findings". Yet, the main text gives the impression that the frequency dependent optical conductivity reported in Fig. 2d is fully supported by the analysis and data presented in this work. Given the authors quoted response, it appears that even the gross features of the optical conductivity, such as the 0.2 eV peak, are not, yet, an experimental fact.

If the authors do not wish to claim to have experimentally determined the frequency dependent optical conductivity at the domain wall, I would feel much more comfortable if that were crystal clear from reading the text. The figure caption should clearly state that the conductivity is coming from a theoretical calculation. Further, adding brief discussion of the specific attributes of the conductivity which will require further study and experimental support would add tremendous clarity for the reader.

If the authors do wish to claim to have experimentally determined specific aspects of the frequency dependent optical conductivity at the domain wall, then more support is needed. From the experimental side if the authors cannot provide hyperspectral linescans with FTIR then frequency dependent nano-imaging data, presented already in Fig. 3 and Fig. 4, can be analyzed where drift and signal to noise issues mentioned by the authors are readily overcome. The available data points of the near field amplitude from imaging data, normalized as $S_{\{DW/BN\}}/S_{\{AB/BN\}}$, can be overlaid on the FTIR data, normalized in the same manner, to fully support proper normalization and null influence of the propagating polariton. The result can be compared to nano-optical spectra calculated with the optical conductivity from the tight binding calculation. Finally, the authors should make crystal clear why gross features that they are

claiming to determine, like the 0.2 eV peak, are absolutely necessary to explain their data.

A few less severe issues are also present in the current version:

I understand that poles of $\text{Im}\{R_p\}$ can be compared to the wavevector of propagating modes and that the high- q frequency dependence of $\text{Im}\{R_p\}$ can bear qualitative resemblance to experimental near-field spectra in certain cases. As is, I am still not sure what specific attributes of the frequency dependent optical conductivity at the domain wall are supported by the comparison of R_p and the data. As the authors claim, "We get a more complete picture by performing simulations that include the tip-sample interactions." So, why does R_p need to be discussed at all?

The colorbars for Fig 1c, d are missing; labels for the signals shown with false color maps are missing throughout the text.

As the near-field amplitude signals in Fig. 2b were normalized to SiO_2 , the units are not arbitrary and the normalization procedure, while mentioned in the rebuttal, is still not mentioned in the text or SOM.

The twist angle of the device in Fig. 3 is missing from the figure caption. The excitation frequency used in Fig 3. d-f is missing.

Reviewer #1

General comment: The authors successfully addressed my concerns. I would like to suggest to publish the manuscript.

Our response: We thank the reviewer for approving our changes and recommending publication of our manuscript.

Reviewer #2

General comment: The authors have successfully addressed all my concerns and comments. I can now recommend it for publication.

Our response: We thank the reviewer for approving our changes and recommending publication of our manuscript.

Reviewer #3

General comment: The authors responses are not completely satisfactory. I do not see any issues with most of their findings, specifically the very nice observation of hBN polaritons reflecting off of the domain-walls in TBG and the assignment that the latter reflection is caused by the spatially inhomogeneous optical conductivity at the domain wall. I emphasize, again, that I believe the latter observation is both interesting and important for the community. I, however, have remaining concerns with the presentation of the key scientific messages and feel that significant revisions are still needed.

Our response: We again thank the reviewer for acknowledging the importance of our work. We respond to the remaining concerns on our presentation below.

Comment 1: The authors have stated that the moiré pattern has been imaged in a fully encapsulated sample with minimal doping without the aid of polaritons in the superstrate within their own work, see Fig 1d. So, the statement that <4 nm hBN superstrates and highly doped samples are needed to visualize the moiré in encapsulated devices without the hBN polaritons appears to be disproved in the same report where it is claimed. If I have accurately assessed the situation, I believe these claims should be modified.

Our response: We thank the reviewer for pointing out the confusion here in our manuscript. We did not properly emphasize that the device used in Fig. 1 “is” a highly doped sample, which was subjected to an oxygen plasma treatment to enhance the plasmonic response. This should have been noted more clearly in the manuscript. (We explicitly added a note in both main text and the method section to address this oversight.) The thickness of the top h-BN layer of this device is 4nm and there is no bottom h-BN layer. Importantly, this device was fabricated to provide a vehicle for *directly* comparing graphene plasmon imaging and h-BN phonon-polariton imaging. The device shown in Fig. 2a on the other hand was the fully encapsulated device with minimal doping, used for later investigation.

Comment 2: *My major technical concern is: it is still not clear if the specific form of the frequency dependent optical conductivity reported in Fig. 2d is fully supported by the data in this report. In the rebuttal the authors have stated: “The resonance at 0.2 eV alone is not essential to explain our findings”. Yet, the main text gives the impression that the frequency dependent optical conductivity reported in Fig. 2d is fully supported by the analysis and data presented in this work. Given the authors quoted response, it appears that even the gross features of the optical conductivity, such as the 0.2 eV peak, are not, yet, an experimental fact.*

If the authors do not wish to claim to have experimentally determined the frequency dependent optical conductivity at the domain wall, I would feel much more comfortable if that were crystal clear from reading the text. The figure caption should clearly state that the conductivity is coming from a theoretical calculation. Further, adding brief discussion of the specific attributes of the conductivity which will require further study and experimental support would add tremendous clarity for the reader.

If the authors do wish to claim to have experimentally determined specific aspects of the frequency dependent optical conductivity at the domain wall, then more support is needed. From the experimental side if the authors cannot provide hyperspectral linescans with FTIR then frequency dependent nano-imaging data, presented already in Fig. 3 and Fig. 4, can be analyzed where drift and signal to noise issues mentioned by the authors are readily overcome. The available data points of the near field amplitude from imaging data, normalized as $S_{\{DW/BN\}}/S_{\{AB/BN\}}$, can be overlaid on the FTIR data, normalized in the same manner, to fully support proper normalization and null influence of the propagating polariton. The result can be compared to nano-optical spectra calculated with the optical conductivity from the tight binding calculation. Finally, the authors should make crystal clear why gross features that they are claiming to determine, like the 0.2 eV peak, are absolutely necessary to explain their data.

Our response: As we already stated in the Line 135-136, Fig. 2d shows a *calculated* local conductivity using a tight-binding model, *which is a theoretical calculation*. We do not wish to assert that it is an experimentally determined quantity. We now added this statement to the figure caption for clarity. We also added the text, “*We note that a complete detailed understanding of the frequency dependent optical conductivity at the domain wall requires further study and experimental support.*” in the revised manuscript.

Comment 3: *I understand that poles of $Im\{R_p\}$ can be compared to the wavevector of propagating modes and that the high-q frequency dependence of $Im\{R_p\}$ can bear qualitative resemblance to experimental near-field spectra in certain cases. As is, I am still not sure what specific attributes of the frequency dependent optical conductivity at the domain wall are supported by the comparison of R_p and the data. As the authors claim, “We get a more complete picture by performing simulations that include the tip-sample interactions.” So, why does R_p need to be discussed at all?*

Our response: We concede this point to the reviewer and have removed discussion from the main document and moved this into Supplementary Information to further support our nano-FTIR measurement.

Comment 4: *The colorbars for Fig 1c, d are missing; labels for the signals shown with false color maps are missing throughout the text.*

Our response: Fig.1c is a TEM image which has typically no colorbar. Fig.1d and Fig 1e share the colorbar on the right. We added a statement in the figure caption of the revised manuscript to mention that Fig.1d and e are shown with false color.

Comment 5: As the near-field amplitude signals in Fig. 2b were normalized to SiO₂, the units are not arbitrary and the normalization procedure, while mentioned in the rebuttal, is still not mentioned in the text or SOM.

Our response: Following other literature (e.g. Ref.18), we deleted the arbitrary unit in the label and mentioned the normalization procedure in the figure caption.

Comment 6: The twist angle of the device in Fig. 3 is missing from the figure caption. The excitation frequency used in Fig 3. d-f is missing.

Our response: The twist angle of the device in Fig. 3a-c is $\sim 0.05^\circ$. The excitation frequency is 1560 cm^{-1} as now mentioned in the main text. We added them to the figure caption in the revised manuscript.